# Factors Influencing Protective Behaviors for Dental Radiation Exposure among Female Korean Dental Hygienists Using Health Belief Model

**DOI:** 10.3390/ijerph19010518

**Published:** 2022-01-04

**Authors:** Su-Yeon Hwang, Jung-Eun Park, Jong-Hwa Jang

**Affiliations:** 1Research Institute for Future Medical Science, Chungnam National University Sejong Hospital, Sejong 30099, Korea; hsyen@naver.com; 2Department of Dental Hygiene, College of Health Science, Dankook University, Cheonan-si 31116, Korea; jepark@dankook.ac.kr

**Keywords:** radiation protection, dental hygienists, health belief model

## Abstract

This study aimed to identify the associated factors for protective behaviors for dental radiation exposure (PBDRE) among dental hygienists using the health belief model (HBM). The HBM, which is composed of perceived susceptibility, perceived seriousness, perceived benefits, perceived barriers, and cues to action, explains preventive behavior. In this study, self-efficacy and modifying factors were additionally applied to the HBM. The subjects of the study were 204 dental hygienists who were working at hospitals or clinics in Korea. An online survey was conducted to measure PBDRE-related factors based on the HBM and self-efficacy. The collected data were analyzed using frequency analysis, *t*-tests, ANOVA, Pearson’s correlation analysis, and hierarchical multiple regression analysis. Regarding modifying factors, performance was found to be high when protection facilities were sufficient (β = 0.24, *p* < 0.001) and low when radiation education was not received (β = −0.16, *p* < 0.05). Among the HBM factors, cues to action for PBDRE (β = 0.28, *p* < 0.001) was the most influential factor in the performance of PBDRE, and the effect of its perceived benefits on radiation exposure was also high (β = 0.17, *p* < 0.001). Regarding the performance of PBDRE according to the HBM, providing education programs on protection can stimulate appropriate cues to action to perform PBDRE. In addition, if the benefits of PBDRE are highlighted, the performance of PBDRE by dental hygienists is increased.

## 1. Introduction

With the development of technologies using X-rays, radiography has become an important diagnostic method [1]. It is used in all healthcare sectors and contributes to health promotion. However, certain amounts of radiation are inevitably delivered to patients, and workers are exposed to occupational radiation [2]. In Korea, dental radiography is performed by dentists or dental hygienists and radiologists under the supervision of a dentist. In most dental institutions, dental hygienists are responsible for radiography and radiation management [3].

Dental radiation is frequently used during oral examinations and for clinical treatment [4]. It is used in the diagnosis, treatment, and monitoring of treatment progress in dental patients [5,6,7]. However, the ionizing radiation needed for tooth and surrounding bone imaging is associated with health risks. Despite low doses, radiation exposure (RE) has been reported to cause thyroid cancer and brain tumors [8]. Therefore, safety management of RE is very important.

The health belief model (HBM) is being applied to explain preventive health behaviors [9]. Under this model, if the subjects/participants are aware of their susceptibility to radiation, they are more likely to take protective actions based on their perceptions. In addition, even those who do not take such actions for their health will take a protective action if they perceive the seriousness of a disease that they may develop. Furthermore, when there are appropriate cues to action, the probability of implementing the action increases.

Perceived benefits increase the likelihood of preventive actions when health behaviors are deemed possible and effective. Moreover, if perceived barriers are greater, the likelihood of protective behavior is decreased. Self-efficacy implies that an individual is willing to perform the health behaviors necessary to achieve the desired results [10,11]. The HBM is an influencing factor in changing and maintaining behavior. Research on health behavior using the HBM has been applied in infection control practice, hepatitis B vaccination uptake, oral health education, and influenza vaccination intention [11,12,13,14].

As the number of radiographic services increases, research on protection against RE continues. Research findings related to protective behavior against radiation exposure have shown that a lack of knowledge and awareness of radiation protection increases the risk of RE. In addition, despite high awareness of radiation protection, low knowledge scores have been reported to result in low levels of protection behaviors [15]. It is essential to perform a careful analysis of the factors influencing protective behaviors for dental radiation exposure (PBDRE) to address the health problems of professionals dealing with radiation. Most of the existing studies of professionals dealing with radiation have been conducted with radiographers and nurses. Research on dental radiology and dental hygienists is limited. Most studies to date have been conducted on behavior; for example, a survey that investigated the degree of acceptance of protection guidelines according to the radiation policy used by American dental hygienists [16] and a study that investigated the radiation safety practices of American dental hygienists [17]. However, comprehensive studies related to the radiation protection behavior of dental hygienists are still lacking.

The hypothesis of this study is as follows: PBDRE will be affected by modifying factors, the HBM, and self-efficacy. Therefore, this study aimed to investigate dental hygienists’ perceived susceptibility, seriousness, benefits, barriers, cues to action, and self-efficacy with respect to dental radiation using the HBM and to determine the degree of performance of PBDRE, in order to understand their interrelationships. In addition, we analyzed the factors affecting the performance of PBDRE.

## 2. Materials and Methods

### 2.1. Study Design and Ethical Consideration

A cross-sectional study was conducted among clinical dental hygienists in Korea. Our study complied with the guidelines of the Declaration of Helsinki and ethical approval for this study was obtained from the institutional review board of Dankook University (DKU 2019-05-021).

### 2.2. Participants and Data Collection

The recruitment criteria for the current study participants included Korean dental hygienists who performed dental radiology work in dental hospitals and clinics in Korea.

According to a national survey on health and medical personnel by the Ministry of Health and Welfare of Korea in 2017, there were 74,589 (99.2%) female registered dental hygienists, and the number of male dental hygienists was found to be 575 (0.77%) [18]. As dental hygienists are a group of medical technicians with an overwhelmingly high proportion of women, the subjects of this study included women only. The sample size was calculated using G*power 3.1 software [19]. With an effect size of 0.15, a *p*-value of 0.05, power of 95%, and the number of predictors at 16, the required sample size was 204 [20]. Of the 210 questionnaires returned, 6 were excluded due to insufficient responses. Finally, 204 questionnaires were used for the analysis.

The study data were collected through an online questionnaire over 2 months from 1 November 2019 (https://docs.google.com/forms/d/e/1FAIpQLScht2tRbzK9SlLJbbuhezzoR3kxvKa3m9n_r6u2luyYhIrr4w/viewform?usp=sf_link, accessed on 20 December 2021). Participants understood the study purpose and consented to participate.

### 2.3. Variables

#### 2.3.1. Modifying Factors

Modifying factors including age, marital status, education level, smoking, drinking, radiation exposure, health impact from radiation exposure, radiation protection equipment effect, radiation protection facilities, and radiation protection education were investigated. Marital status was classified into married and single, and education level was classified into college and graduate school or higher. The frequency of smoking was classified into the following: (1) every day, (2) occasionally, (3) in the past but not now, and (4) not at all. Answers (1) and (2) were finally defined as smoking and (3) and (4) as non-smoking. Alcohol consumption was classified into the following: (1) not at all in the past year, (2) about once a month, (3) about 2–4 times a month, (4) about 2–3 times a week, and (5) 4 times a week or more. Finally, answer (1) was defined as non-drinking and (2)–(5) were defined as drinking. Radiation exposure time was classified as <5 times/day, 6–10 times/day, 11–15 times/day, and ≥16 times/day. The health impact from radiation exposure and radiation protection equipment effects were classified into very likely, moderate, and not at all. Radiation protection facilities were classified into sufficient, moderate, and insufficient, and protection education was recorded as yes or no.

#### 2.3.2. HBM Assessment

The HBM was adapted by modifying and supplementing the tool developed by Rosenstock et al. [21]. It contains a total of 24 questions consisting of perceived susceptibility, perceived seriousness, perceived benefits, perceived barriers, and cues to action, using a 5-point Likert scale (1 point: strongly disagree; 2 points: disagree; 3 points: average; 4 points: agree; 5 points: strongly agree). The reliability of the tool was found to have Cronbach’s α = 0.633 for perceived susceptibility, Cronbach’s α = 0.977 for perceived seriousness, Cronbach’s α = 0.758 for perceived benefits, Cronbach’s α = 0.665 for perceived barriers, and Cronbach’s α = 0.749 for cues to action. The theoretical framework of this study is show in Figure 1.

#### 2.3.3. Self-Efficacy Assessment

The self-efficacy tool was adapted by modifying and supplementing the tool developed by Sherer et al. [22], based on the study by Kim et al. [23]. It consists of 17 questions using a 5-point Likert scale. Each question had a scale with “strongly agree” as 5 points, “agree” as 4 points, “average” as 3 points, “disagree” as 2 points, and “strongly disagree” as 1 point, that is to say, the higher the score, the higher the perception of self-efficacy. The reliability of the tool had Cronbach’s α = 0.809.

#### 2.3.4. Dependent Variable Assessment

The assessment of the performance of PBDRE was adapted by modifying and supplementing the tool developed by Han et al. [24,25]. Each question used a 5-point Likert scale with “strongly agree” as 5 points, “agree” as 4 points, “average” as 3 points, “disagree” as 2 points, and “strongly disagree” as 1 point, indicating that the higher the score, the higher the performance of dental radiation protection behavior. In Han’s study [24], the reliability had Cronbach’s α = 0.851. The reliability in this study was Cronbach’s α = 0.792.

### 2.4. Statistical Analysis

The collected data were analyzed using SPSS software v.20 (SPSS Inc., IBM, Chicago, IL, USA). After confirming that the performance of PBDRE according to the subjects’ general characteristics and radiation-related job characteristics was normally distributed, it was analyzed by independent t-test or one-way ANOVA. After one-way ANOVA analysis, a post hoc analysis was performed on variables showing a significant difference. Pearson correlation was used to investigate correlations between the HBM components. The analysis of factors influencing the performance of PBDRE was performed by hierarchical multiple regression analysis. The significance level was considered to be α = 0.05.

## 3. Results

### 3.1. PBDRE According to General Characteristics

The results for the performance of PBDRE according to the general characteristics of the study subjects are shown in Table 1. Performance was significantly higher in subjects ≥40 years old than in those in their 20s and 30s (F = 4.36, *p* = 0.014), and among those whose education level was graduate school and above (3.34 ± 0.83) (t = −2.60, *p* = 0.016). In addition, PBDRE was significantly higher when radiation affected health (2.76 ± 0.78) than when it did not have much health effect (3.10 ± 0.84) (F = 3.20, *p* = 0.043). It was also significantly higher when the radiation protection facilities were sufficient (3.49 ± 0.75) than when they were insufficient (2.45 ± 0.59) (F = 30.04, *p* < 0.001). Finally, PBDRE was significantly higher in those who had received radiation education (3.38 ± 0.78) than in those who had not (2.65 ± 0.73) (t = 6.63, *p* < 0.001).

### 3.2. Component Distribution of HBM

The component distribution of the HBM is shown in Table 2. The mean of the perceived susceptibility to RE was found to be 3.62 ± 0.65; perceived seriousness to RE was 3.76 ± 0.99; mean of perceived benefits for PBDRE was 3.75 ± 0.66; mean of perceived barriers for PBDRE was 3.41 ± 0.88; cues to action 2.78 ± 0.95; PBDRE, 2.93 ± 0.83; self-efficacy, 2.66 ± 0.48.

### 3.3. Correlations among Protective Behaviors for Radiation Exposure Factors of HBM

Pearson correlation analysis was performed to identify the correlations between the main variables of this study (Table 3). As a result, perceived susceptibility to RE showed a significantly positive correlation with perceived seriousness of RE (r = 0.66, *p* < 0.001), perceived benefits (r = 0.15, *p* < 0.05), and perceived barriers (r = 0.36, *p* < 0.001), while it showed a negative correlation with PBDRE (r = −0.18, *p* < 0.001). Perceived seriousness showed positive correlations with perceived benefits (r = 0.15, *p* < 0.05) and perceived barriers (r = 0.40, *p* < 0.001) but a negative correlation with PBDRE (r = −0.17, *p* < 0.05). Perceived benefits were positively correlated with cues to action (r = 0.22, *p* < 0.001) and PBDRE (r = 0.28, *p* < 0.001), while perceived barriers were positively correlated with self-efficacy (r = 0.21, *p* < 0.001) and negatively correlated with cues to action (r = −0.33, *p* < 0.001) and PBDRE (r = −0.42, *p* < 0.001). Cues to action were positively correlated with PBDRE (r = 0.52, *p* < 0.001).

### 3.4. Hierarchical Multiple Regression Analysis of Protective Behaviors for Radiation Exposure

Hierarchical regression analysis was conducted to determine the effect of applying the HBM on the performance of PBDRE (Table 4). Model 1 (F = 13.24, *p* < 0.001), model 2 (F = 12.69, *p* < 0.001), and model 3 (F = 11.50, *p* < 0.001) were found to be statistically significant. The explanatory power of the regression model was found to be 35% (adjusted R^2^ = 33%) in the first stage, 42% in the second stage (adjusted R^2^ = 39%), and 46% in the third stage (adjusted R^2^ = 42%). Meanwhile, the Durbin–Watson statistic was 1.95, which is close to 2, suggesting that there is no problem regarding the assumption of residual independence. The variance inflation factor (VIF) was also found to be less than 10, indicating no multicollinearity. The result of the benefit test of the regression coefficient showed that model 1 had an effect on the performance of PBDRE in subjects in their 40s (β = 0.13, *p* = 0.029) than in those in the 20s. It was also found to be significant for radiation protection facilities (β = 0.42, *p* < 0.001) and among those who had received education (β = −0.30, *p* < 0.01). In other words, the performance increased when the radiation protection facilities were sufficient in the workplace, and the performance decreased when radiation education was not received.

In model 2, factors related to PBDRE were found to affect subjects in their 40s (β = 0.14, *p* < 0.017) rather than those in their 20s. The PBDRE increased when the protection facilities were better compared with when they were insufficient (β = 0.31, *p* < 0.001), and decreased when radiation education was not received (β = −0.19, *p* < 0.001). Among the HBM factors, cues to action (β = 0.32, *p* < 0.001) resulted in high PBDRE. In model 3, PBDRE was found to be high when protection facilities were sufficient (β = 0.24, *p* < 0.001), and low when radiation education was not received (β = −0.17, *p* = 0.010). Among the HBM factors, cues to action (β = 0.28, *p* < 0.001) was the most influential factor in the performance of PBDRE, and the effect of perceived benefits was also high (β = 0.17, *p* = 0.003).

## 4. Discussion

This study, which is the first to apply the HBM, explored factors affecting the performance of PBDRE for female dental hygienists in Korea. In addition, this study conducted an investigation into developing a plan to improve the performance of PBDRE. Based on the results, the following points were found to merit discussion.

Differences in the performance of PBDRE according to the general characteristics were analyzed. There were significant differences in the performance of PBDRE in terms of age, education level, radiation health effects, radiation facilities, and radiation education. The performance of PBDRE was found to be higher in those in their 40s than those in their 20s and 30s. This is similar to a study involving dental hygiene students and dental hygienists [25]. It is thought that dental hygienists in their 40s or older might have high clinical experience and more educational experience than those in their 20s and 30s. Therefore, it is considered that PBDRE were more ideally performed by subjects ≥40 years old. In addition, the higher the education level, the higher the PBDRE. This is similar to a previous study on 184 nurses in the operating room [26]. The higher the educational background, the greater the chances of academic society and school education, resulting in higher PBDRE. In the case of those who had radiation education, the PBDRE was significantly higher than in those who had not. This was similar to the results of the study examining the difference in the behavior of radiation safety management depending on the availability of education for dental hygienists [27]. It was also consistent with a study on nurses [28]. The performance of PBDRE was significantly higher when they believed that RE had no effect on health. It is considered that they were less concerned about health effects since they were currently performing PBDRE well.

This study identified factors influencing the performance of PBDRE using the HBM. Factors influencing PBDRE were radiation protection facilities, safety education, cues to action, and perceived benefits. Of these variables, cues to action had the greatest impact.

When the radiation protection facilities were superior compared to when they were insufficient (β = 0.24, *p* < 0.001), the performance of PBDRE was affected. This is consistent with previous research in that radiation protection facilities have emerged as a major variable in the performance of PBDRE [29]. Therefore, where radiation protection facilities are insufficient, the performance of PBDRE will increase as more facilities and protective equipment are supplemented. Performance was shown to be lower when no education was received than when it was received (β = −0.17, *p* < 0.05). It is predicted that if radiation education is not received, workers will not be aware of PBDRE [30,31,32]. Therefore, in order to improve PBDRE, dental hygienists must participate in safety education, and it is necessary to strengthen educational participation systematically. Among the HBM factors, cues to action (β = 0.28, *p* < 0.001) and perceived benefits of PBDRE (β = 0.17, *p* < 0.001) had an effect. This is consistent with a study on radiographers [33]. In order to recognize cues to action, a radiation badge is required to be worn at the dental clinic and the RE should be monitored quarterly. The study also suggests that regular training and maximum awareness of cues to action are most effective in changing behavior. As a result, it is expected that the performance of PBDRE will be increased. Perceived benefits were a factor that influenced the performance of PBDRE, applying the likelihood of action. Among the items, minimizing the RE time, measuring perceived benefits, and complying with the radiation management guidelines in the working hospital were found to increase the performance of PBDRE. In addition, if PBDRE is not performed to alleviate RE, the benefits of PBDRE will be recognized by reminding dental hygienists of the negative impact RE may have on their health.

This study has some limitations. Since the study collected data only from Korean female dental hygienists working in some areas, the results can only be generalized to a limited extent. In addition, since the survey was conducted online, the research subjects may be limited to Internet users.

People’s health behavior is achieved through the interaction of various environmental and influencing factors. Therefore, it is difficult to generalize the radiation-related health behaviors of the study subjects using only the HBM applied in this study.

Since there have been few previous studies that have investigated the effects of PBDRE on dental hygienists, and no studies have applied the HBM, this study was discussed in comparison with previous studies on medical workers. Therefore, many more studies using the HBM on PBDRE are needed for dental hygienists in the future. Nevertheless, this study is the first study to identify the factors associated with the performance of PBDRE by applying the HBM to Korean dental hygienists. In addition, it has the advantage of improving the dental radiation safety education method. It suggests that methods of raising awareness of cues to action and perceived benefits are the most effective in improving the performance of PBDRE.

## 5. Conclusions

This study identified factors related to the performance of PBDRE in dental hygienists according to the HBM. The results showed that the greater the cues to action, the higher the performance of PBDRE. In addition, the greater the perceived benefits, the higher the performance of PBDRE. Therefore, an educational program that can stimulate cues to action to perform protective procedures needs to be implemented. If the benefits of these actions are emphasized, the PBDRE performance of dental hygienists will increase.

## Figures and Tables

**Figure 1 ijerph-19-00518-f001:**
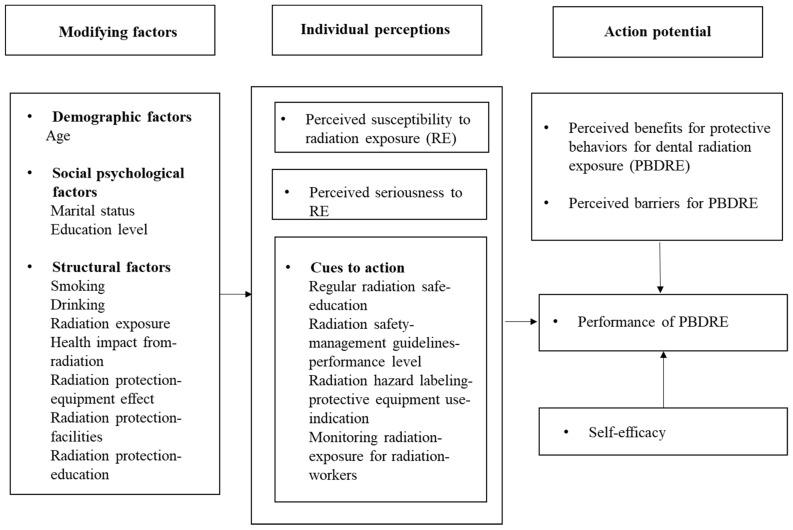
Theoretical framework of study using HBM. RE = radiation exposure; PBDRE = protective behaviors for dental radiation exposure.

**Table 1 ijerph-19-00518-t001:** Differences in PBDRE according to general characteristics by *t*-test or one-way ANOVA.

Characteristics	Category	*n*	Mean	SD	t or F	*p*-Value (scheff)
Age (in years)	20~29 ^a^	130	2.87	0.79	4.36	0.014 ^a,b^ < ^c^
	30~39 ^a^	64	2.94	0.78		
	≥40 ^b^	9	3.70	1.28		
Marital status	Unmarried	149	2.87	0.79	−1.84	0.067
	Married	54	3.11	0.91		
Education level	College	179	2.88	0.81	−2.60	0.016
	≥Graduate school	24	3.34	0.83		
Smoking	No	200	2.93	0.82	0.74	0.459
	Yes	3	3.28	1.14		
Drinking	No	27	2.79	1.00	−0.97	0.332
	Yes	176	2.95	0.80		
Radiation exposure (times/day)	<5 times/day	56	2.94	0.88	1.89	0.133
	6~10 times/day	94	3.05	0.75		
	11~15 times/day	35	2.74	0.82		
	≥16 times/day	18	2.66	0.96		
Health impact from radiation exposure	Very likely ^a^	81	2.76	0.78	3.20	0.043 ^c^ < ^b^ < ^a^
	Moderate ^b^	79	3.01	0.84		
	Not at all ^c^	43	3.10	0.84		
Radiation protection- equipment effect	Very likely	135	2.97	0.80	2.36	0.097
	Moderate	61	2.93	0.87		
	Not at all	7	2.27	0.80		
Radiation protection facilities	Sufficient ^a^	58	3.49	0.75	30.04	<0.001 ^c^ < ^b^ < ^a^
	Moderate ^b^	86	2.88	0.79		
	Insufficient ^c^	59	2.45	0.59		
Radiation protection education	Yes	78	3.38	0.78	6.63	<0.001
	No	125	2.65	0.73		

The *p*-value was derived using independent *t*-test or one-way ANOVA; SD = standard deviation; ^a–c^ represent statistically significant differences (scheff) at α = 0.05; PBDRE = protective behaviors for dental radiation exposure.

**Table 2 ijerph-19-00518-t002:** Component distribution of health belief model.

Characteristics	Range	Min	Max	Mean ± SD
Perceived susceptibility to RE	1–5	1.50	5.00	3.62 ± 0.65
Perceived seriousness of RE	1–5	1.00	5.00	3.76 ± 0.99
Perceived benefits of PBDRE	1–5	2.00	5.00	3.75 ± 0.66
Perceived barriers to PBDRE	1–5	1.00	5.00	3.41 ± 0.88
Cues to action to PBDRE	1–5	1.00	5.00	2.78 ± 0.95
PBDRE	1–5	1.00	5.00	2.93 ± 0.83
Self-efficacy	1–5	1.29	4.24	2.66 ± 0.48

Min = minimum; Max = maximum; SD = standard deviation; RE = radiation exposure; PBDRE = protective behaviors for dental radiation exposure.

**Table 3 ijerph-19-00518-t003:** Correlation of PBDRE-related factors based on HBM.

Characteristics	1	2	3	4	5	6	7
1. Perceived susceptibility to RE	1						
2. Perceived seriousness of RE	0.66 **	1					
3. Perceived benefits of PBDRE	0.15 *	0.15 *	1				
4. Perceived barriers to PBDRE	0.36 **	0.40 **	−0.03	1			
5. Cues to action to PBDRE	−0.18	−0.13	0.22 **	−0.33 **	1		
6. Self-efficacy	0.07	0.07	−0.12	0.21 **	−0.08	1	
7. PBDRE	−0.18 **	−0.17 *	0.28 **	−0.42 **	0.52 **	0.10	1

* *p* < 0.05, ** *p* < 0.001; *p*-value was derived using Pearson correlation analysis; RE = radiation exposure; PBDRE = protective behaviors for dental radiation exposure.

**Table 4 ijerph-19-00518-t004:** Hierarchical multiple regression analysis of PBDRE.

Characteristics	Model 1				Model 2				Model 3			
	B	β	t	*p*-Value	B	β	t	*p*-Value	B	β	t	*p*-Value
Constant	2.85		16.33	<0.001	2.19		6.58	<0.001	2.20		4.49	<0.001
Age (ref. 20–29)												
30–39	0.03	0.02	0.26	0.798	0.04	0.03	0.43	0.666	0.03	0.02	0.29	0.775
40–49	0.53	0.13	2.19	0.029	0.56	0.14	2.40	0.017	0.41	0.10	1.73	0.084
Education level (ref. College)												
≥Graduate	0.28	0.11	1.78	0.077	0.25	0.10	1.69	0.092	0.18	0.07	1.21	0.228
Radiation protection facilities (ref. Insufficient)												
Sufficient	0.77	0.42	5.35	<0.001	0.57	0.31	3.93	<0.001	0.44	0.24	2.87	0.005
Moderate	0.22	0.13	1.73	0.086	0.19	0.12	1.57	0.119	0.13	0.08	1.03	0.305
Radiation protection education (ref. Yes)												
No	−0.51	−0.30	−4.85	<0.001	−0.31	−0.19	−2.88	0.004	−0.28	−0.17	−2.62	0.010
Health effect (ref. Not at all)												
Very likely	0.00	0.00	0.00	0.997	0.12	0.07	0.77	0.440	0.15	0.09	0.96	0.337
Moderate	0.04	0.02	0.32	0.748	0.09	0.05	0.69	0.488	0.12	0.07	0.95	0.344
Perceptions												
Perceived susceptibility to RE					−0.07	−0.06	−0.73	0.464	−0.08	−0.06	−0.83	0.409
Perceived seriousness of RE					0.01	0.01	0.08	0.938	0.00	0.00	0.00	0.997
Cues to action to PBDRE					0.28	0.32	4.76	<0.001	0.24	0.28	4.16	<0.001
Likelihood of action												
Perceived benefits to PBDRE									0.21	0.17	3.02	0.003
Perceived barriers to PBDRE									−0.11	−0.12	−1.68	0.095
Self-efficacy									−0.09	−0.06	−0.94	0.351
F (*p*)		13.24 (*p* < 0.001)				12.69 (*p* < 0.001)			11.50 (*p* < 0.001)			
R2		0.35				0.42			0.46			
Adjusted R2		0.33				0.39			0.42			

Dependent variable = protective behaviors for dental radiation exposure (PBDRE); *p*-value was derived using multiple regression analysis at α = 0.05; RE = radiation exposure.

## Data Availability

Original data are available upon request to the corresponding author.

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
