# Peer review of "Factors Influencing Protective Behaviors for Dental Radiation Exposure among Female Korean Dental Hygienists Using Health Belief Model"

_ijerph, 2022, doi:10.3390/ijerph19010518_

Round 1

Reviewer 1 Report

This manuscript covers a topic that is generally of interest in the field. However, the presents of results makes it hard to understand the study quickly. My main concern is the abstract, as it does not contain all the necessary information in itself, and reading of the main text ist therefore necessary. This should not be the case, the abstract should briefly explain what HBM is and also either avoid using phrases such as “cue to action” and paraphrase this or explain what is meant by this, as it only becomes clear in the methods section of the manuscript. The same goes for the introduction section, it needs to contain short explanations of those words not commonly used. The manuscript should also undergo English language editing. There are a few typos and grammatical errors, mainly, however, many sentences should be rephrased to become more concise. 

Author Response

Thank you for reviewing our study. We have made every effort to incorporate the corrections you requested into the manuscript. In addition, the revised manuscript was edited by a native English speaker for grammar and conciseness.

1) The content of abstract has been added as follows:

▪ The HBM, which is composed of perceived susceptibility, perceived seriousness, perceived benefits, perceived barriers, and cues to action, explains preventive behavior. In this study, self-efficacy and modifying factors were additionally applied to the HBM.

2) From the 44th line of the Introduction, the explanation of the word has been added as follows:

▪ Under this model, if the subjects/participant are aware of their susceptibility to radiation, they are more likely to take protective actions based on their perceptions. In addition, even those who do not take such actions for their health will take a protective action if they perceive the seriousness of a disease that they may develop. Furthermore, when there are appropriate cues to action, the possibility of implementing the action increases.

 We have made our best efforts to accommodate your recommendations in the revised manuscript. Please let us know in detail if you have any further recommendations for modifications. We would be glad to incorporate any further revisions required. Thank you very much.

Reviewer 2 Report

Overall comments

  • Title: why mentioning that they were Female?
  • 13: is the abbreviation HBM in Palatino Linotype style or other? please correct.
  • 31: “radiation workers” is not an adequate term.

Introduction

  • Very well written.
  • Please add a null hypothesis at the end of the Introduction.

Material and methods

  • Figure title must be above the figure and not under the figure (correct Figure 1…).
  • In the structural factors, authors have included smoking and drinking. However, there is no definition on a smoker VS non-smoker and what about a heavy smoker? It must be more illustrated.
  • The same concept with drinking. Define an “drinker”.

Results

  • Line 121: it is hard to understand the meaning of this sentence: “Performance was significantly higher in subjects 120 in their 40s than in those in their 20s and 30s (F = 4.36, p = 0.014) and among those in 121 graduate school and above (3.34 ± 0.83) (t = -2.60, p = 0.016).”. In the table the three groups are well described: 20 a 30. 30 a 39 and > or equal to 40. However, in the text in this misleading. Please simply the text by better defining each tranche of age.
  • Table 1. From marital status to à radiation exposure. Why is there no statistical comparison if there is a significant difference or not between the groups ?

Discussion

  • Line 186: remove aggressively and put more accurately or ideally because aggressively gives a negative aspect however it is so positive that they apply the PBDRE.

The study is very well-written and the statistical analysis of the data is accurate and correct. However, the above points must be corrected.

Author Response

Thank you for reviewing our study. We have made every effort to incorporate the corrections you requested into the manuscript. In addition, the revised manuscript was edited by a native English speaker for grammar and conciseness.

▪ Participants and data collection has been supplemented as follows:

According to a national survey on health and medical personnel by the Ministry of Health and Welfare of Korea in 2017, women accounted for 74,589 (99.2%) of registered dental hygienists, and the number of male dental hygienists was found to be 575 (0.77%) [18]. As dental hygienists are a group of medical technicians with an overwhelmingly high proportion of women, the subjects of this study only included women.

▪ 13: It’s been revised to Palatino Linotype style.

▪ 31: It’s been revised to “workers exposed to occupational radiation.”

Introduction

ᄋ Very well written.

ᄋ Please add a null hypothesis at the end of the Introduction.

Authors’ response: 

▪ The hypothesis are as follows:

 The hypothesis of this study is as follows: PBDRE are affected by modifying factors, HBM, and self-efficacy.

Material and methods

ᄋ Figure title must be above the figure and not under the figure (correct Figure 1…).

ᄋ In the structural factors, authors have included smoking and drinking. However, there is no definition on a smoker VS non-smoker and what about a heavy smoker? It must be more illustrated.

ᄋ The same concept with drinking. Define an “drinker”.

Authors’ response: 

▪ The variables related to smoking and drinking has been added as follows:

 Modifying factors 

Modifying factors including age, marital status, education level, smoking, drinking, radiation exposure, health impact from radiation exposure, radiation protection equipment effect, radiation protection facilities, and radiation protection education were investigated. Marital status was classified into married and single, and education level was classified into college and graduate school or higher. The frequency of smoking was classified into the following: (1) every day, (2) occasionally, (3) in the past, but not now,(4) not at all. Answers (1) and (2) were finally defined as smoking and (3) and (4) as non-smoking. Alcohol consumption was classified into the following: (1) not at all in the past year, (2) about once a month, (3) about 2–4 times a month, (4) about 2–3 times a week, and (5) 4 times or more a week. Finally, answer (1) was finally defined as non-drinking, and (2)–(5) were drinking. Radiation exposure time was classified as <5, 6–10, 11–15, ≥16. Health impact from radiation exposure and radiation protection equipment effect were classified into very likely, moderate, not at all. Radiation protection facilities were classified into sufficient, moderate, and insufficient, and protection education was recorded as yes or no.

Results

ᄋ Line 121: it is hard to understand the meaning of this sentence: “Performance was significantly higher in subjects 120 in their 40s than in those in their 20s and 30s (F = 4.36, p = 0.014) and among those in 121 graduate school and above (3.34 ± 0.83) (t = -2.60, p = 0.016).”. In the table the three groups are well described: 20 a 30. 30 a 39 and > or equal to 40. However, in the text in this misleading. Please simply the text by better defining each tranche of age.

ᄋ Table 1. From marital status to à radiation exposure. Why is there no statistical comparison if there is a significant difference or not between the groups ?

Authors’ response: 

▪ Line 121: The age group has been revised as the followings: 20-29, 30-39, ≥ 40.

▪ Performance was significantly higher in subjects ≥ 40 years old than in those in their 20s and 30s (F = 4.36, p = 0.014) and among those in graduate school and above (3.34 ± 0.83) (t = -2.60, p = 0.016).

▪ The scheffe result was added to confirm the difference between groups (Table 1).

 Discussion

ᄋ Line 186: remove aggressively and put more accurately or ideally because aggressively gives a negative aspect however it is so positive that they apply the PBDRE.

Authors’ response: 

▪ A sentence has been revised by deleting ‘aggressively’ in the sentence.

Therefore, it is considered that PBDRE were more ideally performed by subjects 40 years old.

The study is very well-written and the statistical analysis of the data is accurate and correct. However, the above points must be corrected.

We have made our best efforts to accommodate your recommendations in the revised manuscript. Please let us know in detail if you have any further recommendations for modifications. We would be glad to incorporate any further revisions required. Thank you very much.

Reviewer 3 Report

Authors have well attempted to study aimed to identify the associated factors of protective behaviors for dental radiation exposure (PBDRE) among dental hygienists using the Health Belief Model (HBM). However, there are some issues need clarification.

Pg 2: 57; “However, studies on dental hygienists who perform dental radiography are rare” – Please give relevant references of similar studies on dental hygienists.

Pg 2: 72; size effect or effect size? Please provide justification of an effect size of 0.15?

Pg 2: 86-89; Grammar correction

Pg 3:100; … 1 point, implying may be changed to …1 point; implying

Pg 3:101; Grammar correction

Pg 3:106; … 1 point, implying may be changed to …1 point; implying

Pg 3:10- 108; Grammar correction

Pg 3:111; …The t-test and analysis of variance were used – Please write the proper names of the statistical tests. Also write how analysis of the distribution of data performed was.

Table 1: Please write - Age ( in years)

Table 1: Names of the statistical tests should be written properly

Pg 6:175: …. of PBDRE for female dental hygienists in Korea. – Was the study conducted only among females? Please mention this fact clearly in the methodology, although it has been mentioned in the limitations.

Discussion: Please include additional limitations such as that pertaining to Health Belief model, Online Questionnaire design

Conclusion: Conclusion statement is unclear.

Author Response

Authors have well attempted to study aimed to identify the associated factors of protective behaviors for dental radiation exposure (PBDRE) among dental hygienists using the Health Belief Model (HBM). However, there are some issues need clarification.

1) Line 57: “However, studies on dental hygienists who perform dental radiography are rare” – Please give relevant references of similar studies on dental hygienists.

Authors’ response:

Thank you for reviewing our study. We have made every effort to incorporate the corrections you requested into the manuscript. In addition, the revised manuscript was edited by a native English speaker for grammar and conciseness.

▪ It has been revised by adding the related reference as follows:

Research on dental radiology and dental hygienists is limited. Most studies were conducted on behavior, such as a survey that investigated the degree of acceptance of protection guidelines according to radiation policy used by American dental hygienists [16], and a study that investigated radiation safety practices of American dental hygienists [17]. However, comprehensive studies related to the radiation protection behavior of dental hygienists are still lacking.

 Line 72:  size effect or effect size? Please provide justification of an effect size of 0.15?

Authors’ response: 

The effect size has been set as 0.15 by referring to the below reference. The reference has been added to the manuscript.

With an effect size of 0.15, P-value of 0.05, power of 95%, and number of predictors as 16, the required sample size was 204 [20].

Line 86-89:  Grammar correction

Authors’ response: 

▪ It’s been revised as follows:

▪ The reliability of the tool was found as Cronbach's α = 0.633 in perceived susceptibility, Cronbach's α = 0.977 in perceived seriousness, Cronbach's α = 0.758 in perceived benefits, Cronbach's α = 0.665 in perceived barriers, and Cronbach's α = 0.749 in cues to action. The theoretical framework of this study is show in Figure 1.

Line 100 … 1 point, implying may be changed to …1 point; implying

Line 101: Gmmar correction

Authors’ response:

▪ It’s been revised as follows:

Each question had a scale, with “strongly agree” as 5 points, “agree” as 4 points, “average” as 3 points, “disagree” as 2 points, and “strongly disagree” as 1 point; that is to say, the higher the score, the higher the perception of self-efficacy. The reliability of the tool was Cronbach’s α = 0.809.

Line 106: .. point, implying may be changed to …1 point; implying

Authors’ response: 

▪ It’s been revised as follows:

The performance of PBDRE was adapted by modifying and supplementing the tool developed by Han et al. [20, 21]. Each question used a 5-point Likert scale, with “strongly agree” as 5 points, “agree” as 4 points, “average” as 3 points, “disagree” as 2 points, and “strongly disagree" as 1 point, indicating that the higher the score, the higher the performance of dental radiation protection behavior. In Han’s study [20], reliability was Cronbach’s α = 0.851. The reliability in this study was Cronbach’s α = 0.792.

Line 108: Grammar correction

Authors’ response: 

▪ It’s been revised as follows:

Performance was significantly higher in subjects  40 years old than in those in their 20s and 30s (F = 4.36, p = 0.014) and among those in graduate school and above (3.34 ± 0.83) (t = -2.60, p = 0.016).

Line 111: …The t-test and analysis of variance were used – Please write the proper names of the statistical tests. Also write how analysis of the distribution of data performed was.

Authors’ response: 

▪ It’s been revised as follows:

After confirming that the performance of PBDRE according to the subjects' general characteristics and radiation-related job characteristics was normally distributed, it was analyzed by independent t-test or one-way ANOVA. After one-way ANOVA analysis, a post-hoc analysis was performed on variables showing a significant difference. Pearson correlation was used for correlation between HBM components.

Table 1: Please write - Age ( in years)

Authors’ response: 

▪ It’s been revised.

Table 1: Names of the statistical tests should be written properly

Authors’ response: 

▪ It’s been revised to “Difference of PBDRE according to general characteristics by T-test or one way ANOVA”

 Line 175: …. of PBDRE for female dental hygienists in Korea. – Was the study conducted only among females? Please mention this fact clearly in the methodology, although it has been mentioned in the limitations.

Authors’ response: 

▪ In the ‘Participants and data collection’ section, the reason for the study targeting Korean female dental hygienists was described as follows:

▪ According to a national survey on health and medical personnel by the Ministry of Health and Welfare of Korea in 2017, women accounted for 74,589 (99.2%) of registered dental hygienists, and the number of male dental hygienists was found to be 575 (0.77%) [18]. As dental hygienists are a group of medical technicians with an overwhelmingly high proportion of women, the subjects of this study only included women.

Discussion: Please include additional limitations such as that pertaining to Health Belief model, Online Questionnaire design

Authors’ response: 

▪ The limitations were described as follows in discussion:

▪ This study has some limitations. Since the study collected data only from Korean female dental hygienists working in some areas, the results can be generalized to a limited extent. In addition, since the survey was conducted online, the research subjects might be limited to internet users. People’s health behavior is achieved through the interaction of various environmental and influencing factors. Therefore, it is difficult to generalize the radiation-related health behaviors of the study subjects only by the HBM applied in this study.

Conclusion: Conclusion statement is unclear.

Authors’ response: 

▪ The conclusion was revised as follows:

 This study identified factors related to the performance of PBDRE in dental hygienists according to the HBM. As a result, the greater the cues to action were, the higher the performance of PBDRE. In addition, the greater the perceived benefits were, the higher the performance of PBDRE. Therefore, an educational program that can stimulate cues to action to perform protective procedures needs to be implemented. If the benefits of these actions are emphasized, the PBDRE performance of dental hygienists will increase.

We have made our best efforts to accommodate your recommendations in the revised manuscript. Please let us know in detail if you have any further recommendations for modifications. We would be glad to incorporate any further revisions required. Thank you very much.
